# Targeted Therapy of B7 Family Checkpoints as an Innovative Approach to Overcome Cancer Therapy Resistance: A Review from Chemotherapy to Immunotherapy

**DOI:** 10.3390/molecules27113545

**Published:** 2022-05-31

**Authors:** Bita Amir Taghavi, Nazila Alizadeh, Hossein Saeedi, Noora Karim Ahangar, Afshin Derakhshani, Khalil Hajiasgharzadeh, Nicola Silvestris, Behzad Baradaran, Oronzo Brunetti

**Affiliations:** 1Immunology Research Center, Tabriz University of Medical Sciences, Tabriz 5165665811, Iran; b.amirtaghavi@gmail.com (B.A.T.); alizadeh_imm@yahoo.com (N.A.); saeedi.hossein1995@gmail.com (H.S.); noora.karimi.ahangar@gmail.com (N.K.A.); hajiasgharzadeh@tbzmed.ac.ir (K.H.); 2Medical Oncology Unit-IRCCS Istituto Tumori “Giovanni Paolo II” of Bari, 70124 Bari, Italy; afshin.derakhshani94@gmail.com; 3Medical Oncology Unit, Department of Human Pathology “G. Barresi”, University of Messina, 98122 Messina, Italy; n.silvestris@unime.it; 4Pharmaceutical Analysis Research Center, Tabriz University of Medical Sciences, Tabriz 516566581, Iran; 5Neurosciences Research Center, Tabriz University of Medical Sciences, Tabriz 5165665811, Iran

**Keywords:** immune checkpoint, B7 family checkpoints, chemoresistance, immunotherapy

## Abstract

It is estimated that there were 18.1 million cancer cases worldwide in 2018, with about 9 million deaths. Proper diagnosis of cancer is essential for its effective treatment because each type of cancer requires a specific treatment procedure. Cancer therapy includes one or more approaches such as surgery, radiotherapy, chemotherapy, and immunotherapy. In recent years, immunotherapy has received much attention and immune checkpoint molecules have been used to treat several cancers. These molecules are involved in regulating the activity of T lymphocytes. Accumulated evidence shows that targeting immune checkpoint regulators like PD-1/PD-L1 and CTLA-4 are significantly useful in treating cancers. According to studies, these molecules also have pivotal roles in the chemoresistance of cancer cells. Considering these findings, the combination of immunotherapy and chemotherapy can help to treat cancer with a more efficient approach. Among immune checkpoint molecules, the B7 family checkpoints have been studied in various cancer types such as breast cancer, myeloma, and lymphoma. In these cancers, they cause the cells to become resistant to the chemotherapeutic agents. Discovering the exact signaling pathways and selective targeting of these checkpoint molecules may provide a promising avenue to overcome cancer development and therapy resistance. Highlights: (1) The development of resistance to cancer chemotherapy or immunotherapy is the main obstacle to improving the outcome of these anti-cancer therapies. (2) Recent investigations have described the involvement of immune checkpoint molecules in the development of cancer therapy resistance. (3) In the present study, the molecular participation of the B7 immune checkpoint family in anticancer therapies has been highlighted. (4) Targeting these immune checkpoint molecules may be considered an efficient approach to overcoming this obstacle.

## 1. Introduction

In 2018, 18.1 million cancer cases were diagnosed worldwide, with 9.6 million deaths. Breast and lung cancers are the most common cancers worldwide (2.1 million), accounting for 11.6% of the total number of newly diagnosed cases in 2018. Colorectal cancer is the third most common cancer, with 1.8 million cases. Lung, prostate, and colorectal cancers are three common cancers in men, and the most common of them is lung cancer. In women, three common cancers are breast, colorectal, and lung, the most common of which is breast cancer. Most cancer deaths are related to lung cancer (1.8 million deaths), colorectal cancer (881,000 deaths), stomach cancer (783,000 deaths), liver cancer (782,000 deaths), and breast cancer (627,000 deaths) [1,2,3]. The development of immunotherapy-based approaches against cancer has increased in recent years [4,5,6]. Unlike radiotherapy and chemotherapy, which indirectly interfere with cell growth and survival, immunotherapy directly targets the disease by increasing anticancer immune responses [5,7].

One of the main characteristics of cancer cells is genomic mutability, which leads to their uncontrolled proliferation and antigen expression through mutations and gene rearrangement, which the immune system can recognize. CD8+ T-cells are immune cells involved in anti-tumor responses. These cells need two signals through the antigen-presenting cells (APCs) to stimulate and act on antigens. The first signal is through antigenic peptides on the major histocompatibility complex (MHC), and the next signal is through co-stimulatory molecules and surface receptors such as CD28 [5]. Tumors can escape immune responses through negative feedback mechanisms. The body has evolved to use these mechanisms to prevent immune pathology. These immune responses include inhibitory cytokines like tumor growth factors (TGF-B) and interleukins (IL-10), a variety of inhibitory cells like regulatory T cell (T-reg) and myeloid-derived suppressor cells (MDSC), and regulatory B-cell (B-reg) and metabolic modulators such as indoleamino2,3-deoxygenase (IDO) as well as inhibitory receptors like cytotoxic T-lymphocyte-associated antigen-4 (CTLA-4) and programmed cell death protein-1 (PD-1). Inhibitory receptors are known as immune checkpoints and, their ligands can be found on a variety of cells [5,8].

Immune checkpoint molecules (IChMs) generate signals to regulate the function of T-lymphocytes [9]. Up to now, nearly 70 membrane proteins have been identified as IChMs; most of them belong to the immunoglobulin superfamily and tumor necrosis factor (TNF) [10,11]. The B7 family molecules that are all human molecules are among the most widely studied groups of human IChMs. This family transmits signals that can stimulate T lymphocytes’ proliferation or differentiation, inhibit immune responses, and induce antigenic tolerance. Interaction of B7 ligands with different receptors can inhibit or stimulate T-lymphocytes-mediated immune responses [12]. These molecules have a significant role in the activity, differentiation, effective functions, survival, and prevention of the immune response of T-cells and are involved in tumor evasion [10,13,14]. In the current review, we summarize some features of the B7 family molecules and highlight whether they are involved in chemoresistance. Most studies in this field relate to several family members, including B7-H1, B7-H3, B7-H4, and B7-H6.

## 2. Different Characters of B7 Family Molecules

### 2.1. Specification of the B7 Family

The molecules of the B7 family are structurally part of the immunoglobulins superfamily. These molecules are transmembrane proteins of type I that have an extracellular N-terminal signal [10]. Despite the relatively low homology of the primary structure of B7 family proteins (19–40%), their second and third structures are very similar. They are characterized by the presence of the extracellular IgV and IgC peptide signal domains transmembrane region and cytoplasmic region. All of these molecules are encoded by separate exons. Cytoplasmic domains of these family members are relatively short and usually contain 19–62 amino acid residues that can be encoded by several exons. These domains include serine and threonine and can be phosphorylated and participate in intracellular signal transmission [10]. To date, eleven members of the B7 family are known; the names of this family, and their ligands, are shown in Table 1 [10].

### 2.2. The Expression of the B7 Family on Tumors

Several ligands from the B7 family have been found on tumor cells in the tumor microenvironment (Figure 1). The B7-H1 and B7-DC expression are increased on the surface of many human tumor cells of different histology and anatomic location simultaneously with the rise of the PD-1 receptor expression on T-lymphocytes [15]. Furthermore, in some cases, the expression of B7-H1 is related to the late stage of cancer and reduction of the lifetime [16]. Expression of the B7-H3 gene is found in several types of cancers such as melanoma [17], glioma [18], lung cancer cells [19], pancreas [20], kidneys [21], intestine [22], ovarian [23], mammary glands [24], and gastric cancer [25]. The relationship between the expression of B7-H3 and clinical-pathological parameters has been found in the types of cancers. However, the molecular mechanism which regulates B7-H3 expression and function in tumors is still unclear. B7-H4 is also expressed on breast cancer cells, renal cancer, ovarian carcinoma, pancreatic cancer, brain cancer, and lung cancer [26,27,28,29]. B7-H5 is expressed on cancer cells, such as pancreatic cancer cells [30]. B7-H6 is expressed on types of human cancer cells such as leukemia, lymphoma, melanoma [31], astrocytomas [32], neuroblastoma [33], gliomas [34], epidermoid cancer of the oral cavity, gastrointestinal cancer [35], breast cancer, and ovarian cancer [36]. It is not expressed on normal tissues, and B7-H6 expression is activated by transforming normal cells into cancer cells. The B7-H7 expression is found in different types of human cancer cells such as gastrointestinal and genitourinary cancers [37].

### 2.3. Clinical Importance of B7 Homolog Expression

The expression of B7 ligands on lymphocytic and non-lymphocytic cells indicates their role in regulating immunity in the central organs of the immune system and peripheral tissues.

#### 2.3.1. B7-H1

B7-H1 ligand acts as an inhibitor during linking to its receptor PD-1. These ligands interact with their receptors and induce apoptosis, decreased T-lymphocyte activation, and cytokine activity [38]. B7-H1 is expressed in lymphoid and non-lymphoid tissues [39]. This ligand is also expressed in the APCs, and interferon γ (INFγ) induces B7-H1 expression on the surface of endothelial and epithelial cells [10]. To date, several immune checkpoint inhibitors have been confirmed by the Food and Drug Administration (FDA), such as PD-L1 inhibitors (atezolizumab, durvalumab, avelumab), which are used to treat cancers by immunotherapy [13,40]. B7-H1 or PD-L1 is the predictive marker of response to immunotherapy that we currently have in the clinic [13,40]. Although B7-H1 is known as an IChM, it can produce chemotherapy-resistant cells [41,42,43]. Overexpression of B7-H1 predicts invasive disease, such as increased disease progression and cancer-related mortality in some types of cancers [44,45,46]. The prominent role of B7-H1 in cancer chemoresistance can be a new mechanism that leads to reduced clinical outcomes in cancer patients with B7-H1 overexpression [10].

#### 2.3.2. B7-H2

B7-H2 is expressed on B cells and macrophages, and it is also expressed on non-lymphoid cells under the action of various inflammatory stimuli. B7-H2 binds to the inducible T-cell co-stimulator (ICOS) receptor and shows a stimulating effect on T lymphocytes [47]. ICOS is detected in the activated T lymphocytes, not in the resting T lymphocytes. Early experiments demonstrate that stimulus signals via ICOS increase CD4+ T lymphocyte proliferation but do not increase IL-2 production [47]. Furthermore, ICOS increases production of cytokines such as IFN-γ, TNF-α, IL-4, IL-5, IL-10. ICOS plays a considerable role in regulating Th1 and Th2 cytokines during infection [48]. ICOSL (ICOS-ligand) blockage causes selective suppression of anti-KHL (the keyhole limpet hemocyanin) IgG responses in patients with systemic lupus erythematosus [49]. These results demonstrate that inhibition of B7-H2 interaction with ICOS can be used to treat autoimmune diseases.

#### 2.3.3. B7-H3

B7-H3 mRNA is found in some tissues, including the heart, liver, placenta, prostate, ovary, pancreas, and intestine [50]. The receptor in which the B7-H3 ligand binds is still unknown. Both human isoforms of B7-H3 (4 Ig B7-H3 and 2 Ig B7-H3) inhibit human CD4+ cell proliferation and reduce cytokine production in response to TCR stimulation [42]. Previous studies found the participation of the B7-H3 molecule in enhancing anti-tumor immunity in mouse experiments, but B7-H3 expression in human cancers typically has an opposite effect, and B7-H3 stimulatory activity is currently seen in several clinical studies [51]. In contrast, an antagonistic correlation between B7-H3 expression and cancer clinical outcomes has been observed in more studies [52,53,54,55]. The overexpression of B7-H3 in colorectal cancer is positively related to the progression of cancer. Decreased T-lymphocytes count in cancer, increases TNFα production, which with increasing the level of soluble B7-H3 can inhibit anti-tumor immunity [55]. The therapies targeted by B7-H3 may include monoclonal antibody MGA271 (enoblituzumab) that reacts with B7-H3 and induces antibody-dependent cytotoxicity (ADCC) against many cancer cells [56]. Monoclonal antibodies can be conjugated with cytotoxic agents to cause cancer cell death. 8H9mAb is a monoclonal antibody used against B7-H3 in conjunction with radioactive iodine and is used for diagnosis and treatment of neuroblastoma [57]. Another promising approach is the use of biospecific antibodies to target T cells. These antibodies contain fragments of 2 mAbs that identify different targets and can be used to detect and treat neuroblastoma patients, therefore exhibiting a doublet specificity [58,59] and enabling design and utilization of the Tcell chimeric antigen receptor (CAR) [60,61]. A promising cancer treatment is a complex treatment that involves the B7-H3 blockage with the simultaneous inhibition of other known IChM [62]. In vivo animal studies found that B7-H3 knockdown tumors exhibited a slower growth rate than the control xenografts. Notably, paclitaxel therapy demonstrated significant anti-tumor efficacy in the mice with B7-H3 knockdown tumors and only a marginal impact in the control group. These tumors were inhibited by PTX treatment. Concerning its molecular mechanism, B7-H3 appears to be one of the critical genes for the regulation of Jak2/Stat3 [63]. Stat3 is a cytoplasmic transcription factor that regulates cell differentiation, proliferation, and survival [64,65]. Stat3 is activated by phosphorylation of a variety of kinases like jak and src, and its high activity predicts drug resistance to chemotherapy. It results in the up-regulation of anti-apoptotic factors Bcl-xL, Mcl-1, Bcl-2, and survival [66,67,68]. It is clear that the downregulation of B7-H3 reduces the phosphorylation of stat3 and jak, and overexpression of B7-H3 activates jak2/stat3 signaling; this explains why the B7-H3 knockdown cells are more susceptible to apoptosis caused by paclitaxel [69,70].

#### 2.3.4. B7-H4

Other names for B7-H4 are B7x, B7S1, VTCN1, and DD-0010. mRNA encoding B7-H4 is found in lymphoid and non-lymphoid tissues, whereas expression of B7-H4 is restricted to normal tissues [26,71]. B7-H4 may be expressed on the surface, cytoplasm, and nuclei of cancer cells [27]. B7-H4 is likely to have at least two receptors (stimulating and inhibiting) expressed in T cells under different conditions. Many studies show that overexpression of B7-H4 on cancer cells has an inhibitory function in immune cells. Therefore, the principles of B7-H4-based therapies and therapeutic agents are directed toward eliminating B7-H4-expressing cancer cells [12].

#### 2.3.5. B7-H5

B7-H5 is generally found in hematopoietic tissues. The overexpression of B7-H5 is observed on the surface of myeloid cells, such as the CD14dimCD16+ circulating and the CD14+ CD16+/− inflammatory monocytes and the lymphoid and myeloid DCs [72]. The overexpression of B7-H5 is found in the placenta, which probably refers to this molecule’s role in the fetal tolerance support system [73]. Expression of B7-H5 can take place on cancer cells, including pancreatic cancer cells [30]. Interestingly, the B7-H5 interaction with the still-unknown molecule on T- and B cells’ surface decreases their activation [74]. These observations demonstrate that T- and B-lymphocytes express receptors that bind to B7-H5 [75]. The antibodies that interact with B7-H5 on the surface of T-cells have demonstrated a decrease in their activation. Recently, the first clinical trials of mAb against B7-H5 (JNJ-61610588) have begun for therapy of late-stage cancers (NCT02671955) [12].

#### 2.3.6. B7-H6

B7-H6, also known as NCR3LG1 [76,77], binds to the NKP30 receptor and activates NK cells, and causes tumor cell lysis. NKP30 binds to B7-H6 via the CDR region [78]. B7-H6 is induced by stimulating inflammatory cytokines like IL-1β and TNFα on the surface of monocytes and inflammatory neutrophils CD14+, and CD16+ [31]. Since B7-H6 is predominantly expressed on cancer cells, it is a good target for cancer treatment [79]. In 2010, monoclonal antibodies 4E55 and 17B103 were produced for binding to the B7-H6 extracellular domain [12].

#### 2.3.7. B7-H7

B7-H7 binds to the cells which express the TMIGD2 molecule (Transmembrane and Immunoglobulin Domain Containing 2). The B7-H7 protein expression has been determined in placenta cells and the epithelial cells of the intestine, kidneys, cholecystitis, and the mammary gland. The expression of this protein is not found in the cells of other organs [37]. B7-H7 is expressed on the immune system cells, especially on the surface of human monocytes and macrophages; also, the expression of B7-H7 is found in many types of human cancer cells. The interaction of CD28H with B7-H7 on APCs stimulates human T-cell proliferation and cytokine production such as IFN-γ, IL-5, IL-10, TNF-α, and IL-17 [80]. On the other hand, B7-H7 inhibits both the proliferation of the CD4+ T-cells and CD8+ T-cells, decreasing cytokine production by T-cells. B7-H7 is widely found in cancer cells, and the expression of this molecule is associated with an adverse prognosis [80]. Thus, B7-H7 is a promising target for the creation of therapeutic agents. It appears that not only does B7-H7 targeted therapy increase the anti-tumor immune reactions, but it also inhibits tumor angiogenesis [37].

## 3. Immune Checkpoints and Chemoresistance

Chemoresistance can cause recurrence and metastasis. It challenges the improvement of clinical outcomes for cancer patients and could be a significant barrier to cancer treatment [81]. Molecular mechanisms of chemoresistance include transporter pumps, oncogenes, tumor suppressor genes, mitochondrial alteration, DNA repair, autophagy, epithelial-mesenchymal transition (EMT), cancer stem cells, and exosomes [82,83,84]. Studies today examine the chemoresistance of IChMs and related signaling pathways leading to drug resistance in cancer cells. According to studies, it seems that B7 family members can cause chemoresistance of cancer cells through Erk, PI3K/AKT, Jak/STAT3, and P38 MAPK signaling pathways [63,85,86,87,88,89,90,91].

### 3.1. Role of B7-H1 in Chemoresistance

#### 3.1.1. B7-H1 and Breast Cancer

In a study in 2010, B7-H1 expression was reduced in MDA-MB-231 (breast cancer cells) by a siRNA, and then apoptosis was increased in the treated cells after doxorubicin treatment. These results indicate that B7-H1 can play an essential role in the chemoresistance of breast cancer cells [92].

#### 3.1.2. B7-H1 and Lymphoma

Silencing of PD-L1 returns drug sensitivity and increases apoptosis of tumor cells. In 2012, a study was done on lymphoma cells (Jurkat), in which PD-L1 was silenced and treated with vector plus cisplatin. Various doses of cisplatin (CDDP) were used in the treatment of lymphoma cells. Morphological changes induced by apoptosis were observed in cells and, apoptosis was increased in PD-L1 silenced cells both in vitro and in vivo. In vivo, RNAi PD-L1 was combined with CDDP, which dramatically inhibited cell proliferation, colony formation, and lymphoma invasion, and increased apoptosis with CDDP treatment and prolonged survival of mice treated in this manner [86,93].

#### 3.1.3. B7-H1 and Myeloma

The expression of B7-H1 in myeloma cells is nearly correlated with drug resistance and cell proliferation. The signal that undergoes through the binding of B7-H1 and PD-1 activates the PI3K/AKT signaling pathway, which is a pro-survival signal and induces chemoresistance in multiple myeloma. In 2016, a study was performed by Ishibask-M et al. that investigated the knocked-down B7-H1 in myeloma cells (MOSTI-1) and treated them with Melphalan. Apoptosis was visibly enhanced in these cells compared to control cells [86]. The reverse signal from the binding of B7-H1 to PD-1 is connected to the PI3K/Akt signaling pathway in myeloma cells. Bcl-2, an anti-apoptotic marker, was upregulated in MOSTI-1 cells treated with PD-1-fc-coupled beads, and it caused downregulation of Fas Ligand (FasL) and Fas-Associated Death Domain protein (FADD). The Bcl-2 is activated by the PI3K/Akt pathway [94], and suppression of Akt induces the expression of Fas ligand (FASLG) [95]. Thus, the B7-H1 molecule reaction with PD-1 on myeloma cells induces drug-induced apoptosis through upregulation of the anti-apoptotic response by activating PI3K/Akt [86].

#### 3.1.4. B7-H1 and Triple-Negative Breast Cancer

The knockdown of B7-H1 enhances the sensitivity of TNBC cells to chemotherapy. One study was done in 2018 by Xiao Sheng Wu et al., who used TNBC cell lines with the expression of B7-H1 (MDA-MB-231). The clustered regularly interspaced short palindromic repeats (CRISPR)/CRISPR-associated protein9 (CASP9) method has been employed for creating a B7-H1 bi-allelic knockout (KO) subclone that was carrying a guide RNA (gRNA) sequence-specific to human B7-H1 exon 3. B7-H1 WT (wild type) and KO MDA-MB-231 were treated with cisplatin. B7-H1 was transfected in B7-H1 KO MDA-MB-231 cells; it was shown that the expression of B7-H1 reduced the drug sensitivity of these cells. Cisplatin induced apoptosis by caspase-3 (CASP-3) activation. When treated with anti-B7-H1 antibody alone or in combination with cisplatin, the treatment of cells with antibody and cisplatin dramatically inhibited tumor growth. Expression of B7-H1 with a specific B7-H1 antibody enhanced breast cancer susceptibility to cisplatin. Overexpression of B7-H1 results in increased Erk1/2 activity in cancer cells through binding to DNA-PKCs. The knockdown of B7-H1 via CRISPR/CASP-9 makes breast tumor cells sensitive to chemotherapy in a cell-context-dependent manner. Targeting B7-H1 with monoclonal antibodies also makes cancer cells susceptible to chemotherapy [87].

#### 3.1.5. B7-H1 and Head and Neck Cell Carcinoma

The molecular mechanism of cisplatin-induced chemoresistance in head and neck cell carcinoma (HNSCC) was studied in 2019. In this study, head and neck carcinoma cell lines were used to investigate the role of PD-L1 along with DNA repair complex (Mer11, Rad50, NBS1: MRN) in cisplatin-induced chemoresistance. The cell lines were treated with siRNA, which knocked down the expression of PD-L1 or Nibrin (NBS1). Then the cells were exposed to cisplatin. Chemo resistant cell lines showed a 50% reduction in survival when treated with siNBS1 and cisplatin and a 65% reduction under treatment with siPD-L1 and cisplatin. Use of both siRNAs and cisplatin reduced survival by 80%. There was thus a synergy between cisplatin and PD-L1 inhibition in HNSCC. It was also found that PD-L1 knockdown induces a decrease in Akt phosphorylation and EGFR expression in chemo resistant cell lines [88].

### 3.2. Role of B7-H3 in Chemoresistance

#### 3.2.1. B7-H3 and Breast Cancer

To study the role of B7-H3 in the sensitivity of breast cancer cells to Paclitaxel (PTX), shRNA was used to make B7-H3 knockdown cells in MDA-MB-231. It has been shown that B7-H3 plays a role in tumor cell resistance to PTX and results in drug resistance in breast cancer cells.

#### 3.2.2. B7-H3 and Mantle Cell Lymphoma

In a study by Zhang et al. in 2015, inhibition of B7-H3 expression in mantle cell lymphoma by Bendamustine (Ben) and Rituximab (R) was suggested as a primary treatment for mantle cell lymphoma [96]. They used RNAi (RNA interfering) technology for silencing B7-H3, in Maver and z138 lymphoma cell lines. The B7-H3 knockdown was found to inhibit tumor proliferation and cell cycle progression and migration and invasion of cancer cells. The silencing of B7-H3 also increases drug-induced CASP-3 activity, thereby leading to enhanced apoptosis and treatment efficacy [97].

#### 3.2.3. B7-H3 and Acute Monocytic Leukemia

In a study performed in 2015 on acute monocytic leukemia (AML) cells, the U937 cell line was transfected with B7-H3 siRNA and treated with first-line drugs (AML M5. IDA. AraC). In this study, the survival rate of the transfected group was decreased more than the control group in combination with chemotherapy. The combination of both drugs and B7-H3 siRNA inhibited cell proliferation more than the single drug group, indicating that silencing of B7-H3 enhances drug-induced cytotoxicity and drug-induced apoptosis through increased CASP-3 activity in vitro. Also, tumor-bearing mice were injected with U937 cells in vivo, which were treated with B7-H3 shRNAs and chemotherapy drugs. This experiment also showed an increase in the chemosensitivity of U937 cells in the xenograft model and the anti-tumor activity [98]. 

#### 3.2.4. B7-H3 and Neuroblastoma

Artemether derivatives from Chinese medicinal herbs are prescribed for malaria patients [99]. In recent years, it has been shown that artemether has potential therapeutic effects on a variety of malignancies [100,101]. It has been demonstrated that artemether and its derivatives have anti-tumor activity. The artemether seriously reduces neuroblastoma cell line proliferation. Moreover, DNA synthesis and cell viability in the neuroblastoma tumor cell lines treated with artemether and doxorubicin were remarkably lower than in cells treated with doxorubicin alone. Artemether increases sensitivity in neuroblastoma cell lines. Doxorubicin suppresses B7-H3 expression in neuroblastoma cell lines and, inhibition of B7-H3 with doxorubicin in cells treated with artemether further suppresses tumor growth. Artemether cannot sensitize neuroblastoma cells to doxorubicin in neuroblastoma cells with B7-H3 overexpression. Thus B7-H3 provides a drug resistance to artemether in neuroblastoma cells [101].

#### 3.2.5. B7-H3 and Melanoma

A study investigated the chemoresistance of B7-H3 and its molecular mechanism in melanoma cells in the year 2019. In this study, melanoma cell lines with decreased B7-H3 expression were treated with dacarbazin (DTIC) and cisplatin drugs in vitro. Mice were also treated with subcutaneous injection of cells with B7-H3 knockdown in vivo. Decreased B7-H3 protein expression reduced colony formation of treated cells with both drugs. Even in mice treated with DTIC drug, tumor growth was reduced in mice injected with B7-H3 knockdown cells. Melanoma cells with low expression of B7-H3 are more sensitive to DTIC and cisplatin. Examination of the molecular basis of drug resistance has shown that B7-H3 expression is involved in P38-MAPK activation. Chemosensitivity increases DUSP10 (dual specificity phosphatase 10) expression. There is an upregulation of DUSP10 mRNA levels in B7-H3 knockdown cells. Overall, the enhanced sensitivity of knockdown B7-H3 cells is eliminated by DUSP10 knockdown with siRNA. Indeed, the molecular mechanism of drug resistance mediated by B7-H3 is associated with reduced DUSP10 expression and, consequently, p38-MAPK activation in melanoma cells [89].

#### 3.2.6. B7-H3 and Ovarian Cancer

Accumulated evidence suggests that B7-H3 expression is connected to tumor growth and prognosis in various types of tumors. In a study conducted in 2019, B7-H3 overexpression was found in patients with advanced-stage ovarian cancer, thus indicating that B7-H3 is wholly associated with the tumor stage in ovarian cancer. Furthermore, an investigation of B7-H3 role in ovarian cancer chemoresistance was performed in A2780, OVCAR3 cell lines with B7-H3 knockdown. The expression of B7-H3 facilitates the proliferation of these cells as well as colony formation, tumorigenesis, and chemoresistance in these cells when treated with cisplatin and Paclitaxel (PTX). B7-H3 overexpressing prevents cisplatin and PTX-induced apoptosis in cells indicates drug resistance induced by B7-H3 in ovarian cancer cells. In vivo, OVCAR3 cells were also subcutaneously injected into mice and then treated with anti-B7-H3 and anti-B7-H4 antibodies. Then cells were treated with PTX and cisplatin, and the growth of tumors was examined. The results showed a decrease in tumor volume and prolonged survival. B7-H3 expression induces the activation of the PI3K/Akt signaling pathway. However, the PI3K/Akt signal is unable to regulate tumor cell apoptosis to chemotherapy [90]. It has been reported that the downstream of Akt protein, Bcl-2, is involved in the anti-apoptotic process of a variety of cancer cells, leading to a decrease in cytotoxicity and drug resistance, B7-H3 upregulates the expression of Bcl-2 and promotes drug resistance [102,103]. 

### 3.3. Role of B7-H4 in Chemoresistance

#### 3.3.1. B7-H4 and Melanoma

Studies have demonstrated that B7-H4 plays a severe role in melanoma cancer progression. B7-H4 overexpression in melanoma patients is associated with poor progression metastasis. On the other hand, it was found that apoptosis is increased in doxorubicin (DOX)-treated cells with downregulated B7-H4. A study conducted by Wang et al. showed that DOX-resistant MDA-MB-435 cells and MDA-MB-435 WT parental cells were treated and assayed with B7-H4 antibodies on different days. MDA-MB-435 cells are originally from the M14 melanoma cell line. This monotherapy was restrained by 40% of parental cell growth and 25% of DOX-resistant cell growth. In the next step, they tested the sensitivity of melanoma cells to DOX, Paclitaxel, and carboplatin. They found that treatment with B7-H4 antibodies could increase the sensitivity of cell lines to these drugs.

#### 3.3.2. B7-H4 and Breast Cancer

Recent studies have shown that B7-H4 activity led to abnormal down-regulation of the Akt pathway in EBV-Positiv-B-cell lymphoma cells [104]. In the study by Wang, the B7-H4 silencing induced upregulating PI3K/Akt signaling pathway in DOX-treated and non-treated cells. Therefore, it seems that B7-H4 neutralizing antibodies vastly reduce tumor cell viability in vitro, and the protein confers resistance to doxorubicin by reducing the sensitivity of breast cancer cells to apoptosis, mediated via the PTEN/PI3K/Akt pathway [91].

### 3.4. Role of B7-H6 in Chemoresistance

#### 3.4.1. B7-H6 and High-Risk Neuroblastoma

High-risk neuroblastoma (HR-NB) is an important malignancy in children. There are no mechanisms for monitoring the immune system in this disease. NK cells play an essential role in the metastasis and survival of HR-NB after Myeloablatiun multimodal chemotherapy and stem cell transplantation through the interaction of the B7-H6 and NKP30 receptors. NB cells expressing the B7-H6 ligand stimulate NK cells in a non-dependent NKP30 manner. The concentration of soluble B7-H6 in serum when the NKP30 is down-regulated correlates with bone marrow metastasis and chemotherapy resistance. The soluble B7-H6 in the serum of HR-NB patients inhibits NK-cell function in vitro [32].

#### 3.4.2. B7-H6 and Non-Hodgkin’s B-Cell Lymphoma

B7-H6 knockdown increases the sensitivity of non-Hodgkin’s B-cell lymphoma tumor cells to chemotherapy. To investigate whether B7-H6 knockdown makes lymphoma cells susceptible to chemotherapeutic drugs, a study was conducted by Feifei Wu in 2010. The B-cell lymphoma cell lines (non-infected cells; CA46) were used in this study, and the expression of B7-H6 was silenced with shRNA (short-hairpin RNA) by Lentivirus-based RNA interference transfection. B7-H6 shRNA (CA46 sh B7-H6) cells were created and the expression of B7-H6 was reduced at mRNA and protein levels. Both groups of CA46 shB7-H6 cells and non-targeted control (CA46sh Ctrl) cells were treated with diverse concentrations of VCR (Vincristine) and Dex (dexamethasone) chemotherapy for 24 h, and cell viability was measured. The cell viability was significantly lower in the CA46-sh-B7-H6 group than in the CA46-sh-control group. In a study, apoptosis induced by VCR and Dex was also investigated. That data showed that apoptosis was increased significantly in CA46 sh B7-H6 cells compared to CA46 cells. It is thought that susceptibility to chemotherapy is mediated through apoptosis-related proteins. The expression of apoptosis proteins like Bcl-2, Bcl-xl, CASP-3, survivin, and c-myc was examined in the CA46sh B7-H6 group treated with a drug versus the CA46 and CA46 shB7-H6 cells non-treated with the drug by Western blotting. According to this study and its results, B7-H6 knockdown makes B-cell lymphoma cells sensitive to chemotherapy [105].

## 4. Conclusions

The members of the B7 family can demonstrate both inhibiting and stimulating properties (Table 1). Nowadays, eleven representatives of the B7 family are known: B7-1 (CD80), B7-2 (CD86), B7-H1 (PD-L1, CD274), B7-DC (PDCD1LG2, PD-L2, CD273), B7-H2 (B7RP1, ICOS-L, CD275), B7-H3 (CD276), B7-H4 (B7x, B7S1, Vtcn1), B7-H5 (VISTA, Platelet receptor Gi24, SISP1), B7-H6 (NCR3LG1), B7-H7 (HHLA2), and ILDR2 [10]. Studies show that B7-H1 is highly expressed in various lymphomas, and its expression contributes to the resistance of these cells to CDDP [92]. It is also associated with invasive myeloma behaviors, such as cell growth and drug resistance induced in the cells by anti-apoptotic responses through the AKT signaling pathway [86]. B7-H1 plays a role in drug resistance in HNSCC cell lines [88]. B7-H3 expression activates the PI3K/AKT signaling pathway and up-regulates Bcl-2 in protein levels, thereby causing chemoresistance in ovarian cancer [90,102,103]. B7-H3 knockdown increases the chemosensitivity of mantle cell lymphoma [97]. Also, this ligand reduces the sensitivity of breast cancer cells to apoptosis and drug resistance through the Jak2/STAT3 signaling pathway [63]. B7-H3 silencing in acute monocytic leukemia inhibits tumor proliferation, cell cycle progression, migration, and invasion, and increases drug-induced apoptosis [98]. B7-H3 knockdown in melanoma cells resistant to DTIC, which is a result of MAPK activation in melanoma cells, led to enhanced sensitivity [89]. In neuroblastoma cells, B7-H3 expression causes these cells to become resistant to doxorubicin [101]. The B7-H4 protein is involved in the development and progression of TNBC and appears to induce drug-resistance via the PTEN/PI3K/AKT signaling pathway [91]. B7-H6 silencing increases the chemosensitivity of B-cell lymphoma cells [105]. These findings offer new insights into the role of B7 ligands in human cancers that could serve as a prognostic biomarker and a promising and new therapeutic target in combination with chemotherapy. Targeting these ligands with antibodies or silencing them with methods such as interfering RNA may be a new approach to sensitizing cancer cells to chemotherapy. In addition, more research should be done on the role of these ligands and other members of the B7 family, such as B7-H7and ILDR2 (new members of this group), on the drug resistance of cancer cells. Furthermore, research is needed on the exact signaling pathways of B7 members that participate in oncogenesis and resistance to chemotherapy.

## Figures and Tables

**Figure 1 molecules-27-03545-f001:**
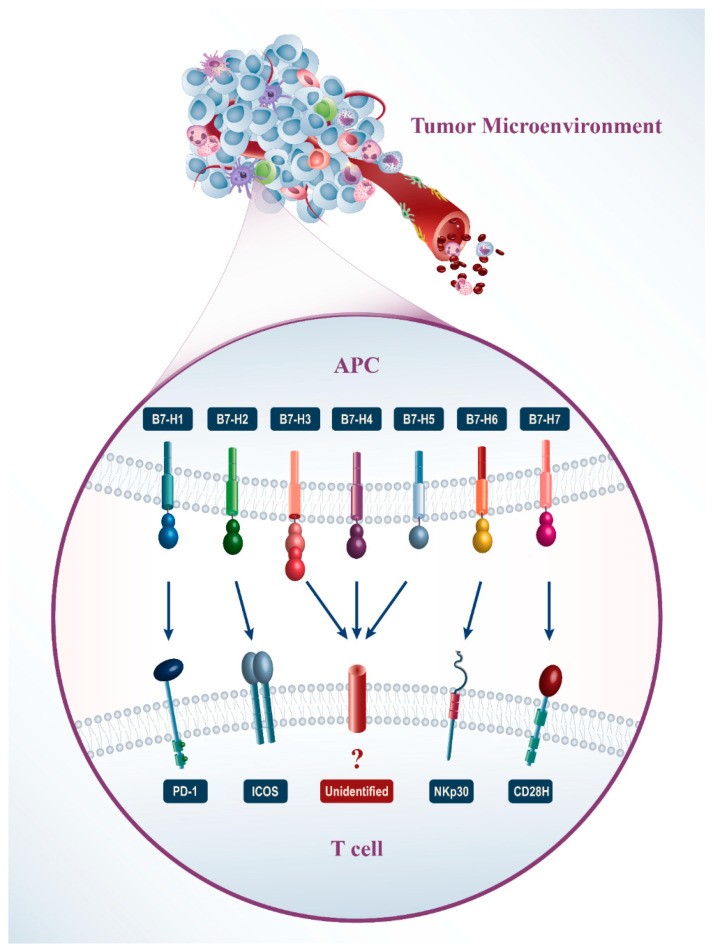
In recent years, immunotherapy has received much attention in cancer therapy and immune checkpoint molecules have a predominant position herein. These molecules are expressed on the surface of several immune cells such as T lymphocytes or other immune cells that are thought to interact with the ligand (e.g., in some cases NK cells, B cells, monocytes) and generate signals to regulate their function. Here, we summarize some features of the B7 family immune checkpoint molecules and highlight whether they are involved in chemoresistance processes and may be considered emerging targets to overcome cancer therapy resistance.

**Table 1 molecules-27-03545-t001:** The members of B7 family checkpoints and their alternative names, ligands, and type of response.

B7 Family Member	Alternative Names	Ligand	Type of Response
**B7-1**	CD80	CTLA-4, CD28	Positive
**B7-2**	CD86	CTLA-4, CD28	Positive
**B7-H1**	PD-L1, CD274	PD-1	Negative
**B7-DC**	PDCD1LG2, PD-L2, CD273	PD-1	Negative
**B7-H2**	B7RP1, ICOS-L, CD275	ICOS	Negative
**B7-H3**	CD276	?	Positive/Negative
**B7-H4**	B7x, B7S1, Vtcn1	?	?
**B7-H5**	VISTA, Platelet receptor, Gi24, SISP1	?	Negative
**B7-H6**	NCR3LG1	NKp30	Negative
**B7-H7**	HHLA2	TMIGD2	Negative/?
**ILDR2**	?	?	Negative/?

## Data Availability

Data sharing is not applicable to this article as no datasets were generated or analyzed during the current study.

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
