# Peer review of "Targeted Therapy of B7 Family Checkpoints as an Innovative Approach to Overcome Cancer Therapy Resistance: A Review from Chemotherapy to Immunotherapy"

_molecules, 2022, doi:10.3390/molecules27113545_

Round 1
Reviewer 1 Report
The review article by Bita Amir Taghavi et al. analyzes the B7 family molecules in the context of anti-tumor therapy through checkpoint inhibitors and their role in regulating T-cell responses. This article summarizes the literature data regarding B7-H1 through B7-H7 molecules, with insights for B7-H1, B7-H3, B7-H4, and B7-H6 regarding chemoresistance in various types of cancers. The article is well written and well articulated and clear in its chapters and subchapters.
The figure and table are clear, however perhaps could be improved.
Therefore, I have no major criticisms of the article, however, some minor considerations.
In the figure the structure not yet identified for B7-H3, B7-H4, and B7-H5, I would distribute it into 3 structures to be identified, because they will probably be 3 distinct structures. Below where the T cell appears it might be helpful to mention perhaps in parentheses also the other immune cells that are thought to interact with the ligand (e.g. in some cases NK cells, B cells, monocyte).
In addition, the B7-H5 molecule is also known as VISTA and not B7-H6. Therefore this should be put in the text and reference No. 74 is misplaced because it turns out to be indicated as B7-H6. (Le Mercier, I.; Chen, W.; Lines, J.L.; Day, M.; Li, J.; Sergent, P.; Noelle, R.J.; Wang, L. VISTA Regulates the Development of 657 Protective Antitumor Immunity. Cancer Res. 2014, 74, 1933–1944, doi:10.1158/0008-5472.CAN-13-1506).
Also since B7 family molecules are all human molecules it should be stated in the text in the introduction.
Also since some stimulations of these checkpoint inhibitors have a stimulatory effect on the T cell, perhaps it would be helpful in the Table that the type of response (positive or negative) is indicated.
Author Response
Dear Editor/Reviewers
Thank you very much for reviewing our manuscript. Your insights led to the improvement of our paper. We have revised the manuscript according to your useful suggestions. Please find below a point-by-point response to comments. We hope our revision has improved the manuscript and will be suitable for publication in the journal of “Molecules”. All of the changes in the manuscript are highlighted in yellow.
Reviewer 1:
Comments and Suggestions for Authors
The review article by Bita Amir Taghavi et al. analyzes the B7 family molecules in the context of anti-tumor therapy through checkpoint inhibitors and their role in regulating T-cell responses. This article summarizes the literature data regarding B7-H1 through B7-H7 molecules, with insights for B7-H1, B7-H3, B7-H4, and B7-H6 regarding chemoresistance in various types of cancers. The article is well written and well articulated and clear in its chapters and subchapters.
The figure and table are clear, however perhaps could be improved.
Therefore, I have no major criticisms of the article, however, some minor considerations.
In the figure the structure not yet identified for B7-H3, B7-H4, and B7-H5, I would distribute it into 3 structures to be identified, because they will probably be 3 distinct structures. Below where the T cell appears it might be helpful to mention perhaps in parentheses also the other immune cells that are thought to interact with the ligand (e.g. in some cases NK cells, B cells, monocyte).
Thank you very much for your positive attitude and detailed comments. In addition to the revision of the text, there have been changes in figure caption to meet your valuable comment:
Figure 1. In recent years, immunotherapy has received much attention in cancer therapy and immune checkpoint molecules have a predominant position herein. These molecules are expressed on the surface of several immune cells such as T lymphocytes or other immune cells that are thought to interact with the ligand (e.g. in some cases NK cells, B cells, monocyte) and generate signals to regulate their function. Here, we summarized some features of the B7 family immune checkpoint molecules and highlighted whether they are involved in chemoresistance processes and may be considered as emerging targets to overcome cancer therapy resistance.
In addition, the B7-H5 molecule is also known as VISTA and not B7-H6. Therefore this should be put in the text and reference No. 74 is misplaced because it turns out to be indicated as B7-H6. (Le Mercier, I.; Chen, W.; Lines, J.L.; Day, M.; Li, J.; Sergent, P.; Noelle, R.J.; Wang, L. VISTA Regulates the Development of 657 Protective Antitumor Immunity. Cancer Res. 2014, 74, 1933–1944, doi:10.1158/0008-5472.CAN-13-1506).
Thank you very much for your detailed comment. We fixed the mistake in references 76 and 77.
- Cherif, B.; Triki, H.; Charfi, S.; Bouzidi, L.; Kridis, W. Ben; Khanfir, A.; Chaabane, K.; Sellami-Boudawara, T.; Rebai, A. Immune Checkpoint Molecules B7-H6 and PD-L1 Co-Pattern the Tumor Inflammatory Microenvironment in Human Breast Cancer. Sci. Rep. 2021, 11, 7550, doi:10.1038/s41598-021-87216-9.
- Jiang, T.; Wu, W.; Zhang, H.; Zhang, X.; Zhang, D.; Wang, Q.; Huang, L.; Wang, Y.; Hang, C. High Expression of B7-H6 in Human Glioma Tissues Promotes Tumor Progression. Oncotarget 2017, 8, 37435–37447, doi:10.18632/oncotarget.16391.
Also since B7 family molecules are all human molecules it should be stated in the text in the introduction.
Thank you very much for your detailed comment. We revised the text according your valuable comments.
The B7 family molecules that are all human molecules are among the most prestigious groups of human IChMs that are widely studied.
Also since some stimulations of these checkpoint inhibitors have a stimulatory effect on the T cell, perhaps it would be helpful in the Table that the type of response (positive or negative) is indicated.
Thank you very much for your detailed comment. In addition to the revision of the text, there have been changes in table 1 to meet your valuable comment:
Table 1. The members of B7 family checkpoints and their alternative names, Type of response, and ligands.

Reviewer 2 Report
This is an interesting and informative review about the targeted therapy of B7 family checkpoints to overcome cancer therapy resistance, however, there are some issues that need to be addressed by the authors in order to improve their manuscript:
1-There are typos and grammatical mistakes throughout the manuscript and it needs to be read and corrected by a native English speaker. For instance, abstract, “there were 18.1 million cancer cases”, etc.
2-Human genes, in vitro and in vivo should be italicized throughout the manuscript. Human proteins should be in the upper cases.
3-Abbreviations should be fully explained at their first occurrence in the text, e.g., PTX line 174.
4-Some paragraphs are way too long and could be divided into multiple and relevant paragraphs to make it easier for the readers to follow and apprehend.
5-Line 179, please replace surviving with survivin.
6-It is strongly recommended that to link immune checkpoints and cancer therapy resistance, the authors start with the clinical evidence and then findings in cell lines.
7-Line 276, the abbreviation for Caspase-3 is CASP-3.
8-Line 299, MDA-MB-435 cells are originally melanoma, not breast cancer. The same with line 374.
9-Line 425, PI3K/AKT is correct.
10-Tables could be used to summarize the major findings. For instance, the role of each family member in therapy resistance in cancer types.
Author Response
Dear Editor/Reviewers
Thank you very much for reviewing our manuscript. Your insights led to the improvement of our paper. We have revised the manuscript according to your useful suggestions. Please find below a point-by-point response to comments. We hope our revision has improved the manuscript and will be suitable for publication in the journal of “Molecules”. All of the changes in the manuscript are highlighted in yellow.
Reviewer 2:
Comments and Suggestions for Authors
This is an interesting and informative review about the targeted therapy of B7 family checkpoints to overcome cancer therapy resistance, however, there are some issues that need to be addressed by the authors in order to improve their manuscript:
Thank you very much for reading our manuscript and for your positive attitude.
1-There are typos and grammatical mistakes throughout the manuscript and it needs to be read and corrected by a native English speaker. For instance, abstract, “there were 18.1 million cancer cases”, etc.
Thank you very much for your detailed comment. We fixed the mistake in abstract section.
It is estimated that there were 18.1 million cancer cases worldwide in 2018
2-Human genes, in vitro and in vivo should be italicized throughout the manuscript. Human proteins should be in the upper cases.
Thank you very much for your detailed comment. We fixed the mistakes.
3-Abbreviations should be fully explained at their first occurrence in the text, e.g., PTX line 174.
Thank you very much for your detailed comment. We revised the abbreviations of the text according to your comment.
4-Some paragraphs are way too long and could be divided into multiple and relevant paragraphs to make it easier for the readers to follow and apprehend.
Thank you very much for your detailed comment. We revised the text according your comments.
5-Line 179, please replace surviving with survivin.
Thank you very much for your detailed comment. We fixed the mistake.
6-It is strongly recommended that to link immune checkpoints and cancer therapy resistance, the authors start with the clinical evidence and then findings in cell lines.
Thank you very much for your detailed comment. We revised the text according your valuable comments.
7-Line 276, the abbreviation for Caspase-3 is CASP-3.
Thank you very much for your detailed comment. We fixed the mistake.
8-Line 299, MDA-MB-435 cells are originally melanoma, not breast cancer. The same with line 374.
Thank you very much for your detailed comment. We revised the section.
9-Line 425, PI3K/AKT is correct.
Thank you very much for your detailed comment. We fixed the mistake.
10-Tables could be used to summarize the major findings. For instance, the role of each family member in therapy resistance in cancer types.
Thank you very much for your detailed comment. In addition to the revision of the text, there have been changes in table 1 to meet your valuable comment.

Round 2
Reviewer 2 Report
The authors have successfully addressed my comments and I have no further comments to raise.